# Perceived built environment, health-related quality of life and health care utilization

**Paco Cerletti**[1,2], **Ikenna C. Eze**[1,2], **Dirk Keidel**[1,2], **Emmanuel Schaffner**[1,2], **Daiana Stolz**[3], **Paola M. Gasche-Soccal**[4], **Thomas Rothe**[5], **Medea Imboden**[1,2], **Nicole Probst-Hensch**[1,2]*

**1** Swiss Tropical and Public Health Institute, Basel, Switzerland, **2** University of Basel, Basel, Switzerland, **3** Clinic for Respiratory Medicine and Pulmonary Cell Research, University Hospital Basel, Basel, Switzerland, **4** Division of Pulmonary Medicine Geneva University Hospitals, Geneva, Switzerland, **5** Department of Internal Medicine and Pneumology, Zuercher Hoehenklinik Davos, Davos, Switzerland

* nicole.probst@swisstph.ch

## Abstract

Previous research has shown that the built environment plays a crucial role for health-related quality of life (HRQoL) and health care utilization. But, there is limited evidence on the independence of this association from lifestyle and social environment. The objective of this cross-sectional study was to investigate these associations, independent of the social environment, physical activity and body mass index (BMI). We used data from the third follow-up of the Swiss study on Air Pollution and Lung and Heart diseases In Adults (SAPALDIA), a population based cohort with associated biobank. Covariate adjusted multiple quantile and polytomous logistic regressions were performed to test associations of variables describing the perceived built environment with HRQoL and health care utilization. Higher HRQoL and less health care utilization were associated with less reported transportation noise annoyance. Higher HRQoL was also associated with greater satisfaction with the living environment and more perceived access to greenspaces. These results were independent of the social environment (living alone and social engagement) and lifestyle (physical activity level and BMI). This study provides further evidence that the built environment should be designed to integrate living and green spaces but separate living and traffic spaces in order to improve health and wellbeing and potentially save health care costs.

## 1. Introduction

The environment, which can range from the natural (greenspaces, lightly populated), built or physical environment (man-made, densely populated) to the social environment (family, peers, community engagement), serves as the context of life, and contributes to its quality in terms of health, well-being and diseases [1,2]. The built environment impacts exposures such as noise, environmental pollutants and general neighborhood conditions including infrastructural adequacy, which can facilitate or hinder physical and psychological functioning [1–6].

Multiple health outcomes including headaches, arthritis and various respiratory morbidities were also associated to the built environment [7,8]. The perception of the built environment seems to affect HRQoL, defined as "how well a person functions in their life and his or her

**Funding:** This research was supported by the Swiss National Science Foundation (grants no

33CS30-177506/1). The study could not have been done without the help of the study participants, technical and administrative support and the medical teams and field workers at the local study sites.

perceived well-being in physical, mental, and social domains of health" [9]. HRQoL is highly correlated with the health status [10,11]. Positive perceptions of neighborhood aesthethics, access to shops, services, public transportation and green spaces were associated with higher HRQoL scores [12,13].

A more integrated approach investigating both, different domains ofthe perceived built environment and individual lifestyle characteristics on HRQoL is critical to the advancement of Public Health policies and urban planning enabling healthy aging for large parts of the population. But the understanding of pathways and mechanisms linking the perceived built environment to HRQoL remains limited. In particular, evidence on the role of the perceived social environment and of physical activity in relation to the built environment remains understudied [14]. Individuals with poor perceptions of social support seem to evolve more aggravated mental health issues with stronger symptoms in disease-outcomes compared to individuals perceiving their social network environment positively, even though reverse causation cannot be excluded in these cases [15,16]. However, whether the association of the perceived built environment with HRQoL is independent of the perception of the social environment is not clear.

Furthermore, physical activity (and related to it obesity) is a priority factor when investigating mechanisms interlinking the built environment and HRQoL, given the rising prevalence of physical activity limitations and associated social, physical, and financial costs in urban and aging populations [17–19]. It is broadly documented that the living environment plays a central role in promoting or inhibiting physical activity [20–22]. In contrast, whether the association of the perceived built environment with HRQoL is independent of physical activity levels remains elusive.

The perception of environmental characteristics might not only influence HRQoL, but also health-seeking behavior [23,24]. From a "Health in All Policy" perspective [25,26], it seems important to show the associations of HRQoL and health care utilization in order to highlight inadequacies related to environmental and social policies. Yet, no studies that we could find have linked single characteristics of the physical environment to health care utilization as a downstream consequence of poor HRQoL [27,28].

In this cross-sectional analysis embedded in the population-based Swiss Cohort Study on Air Pollution and Lung and Heart Diseases in Adults (SAPALDIA) we investigated: (1) the association of the perceived built environment with HRQoL and health care utilization and (2) whether the association was independent of the social environment, physical activity and BMI.

## 2. Methods

### 2.1 Study population

SAPALDIA, initiated in 1991 (SAPALDIA1), is a population-based cohort with associated biobank involving 9'651 adults (18–62 years) drawn from eight representative Swiss areas aimed originally at understanding the respiratory impact of air pollution exposure in the Swiss population [29]. In the subsequent three follow-ups completed over 25 years (SAPALDIA2, 2001/2002, 8'047 participants; SAPALDIA3, 2010/2011, 6'088 participants [30]; and SAPALDIA4, 2017/2018, 5'149 participants) the study expanded into cardio-metabolic outcomes, well-being and healthy aging. The current cross-sectional analysis was performed using SAPALDIA4 data. We included 1980 SAPALDIA4 participants who had complete data on the perceived built and social environment, HRQoL, health care utilization as well as other relevant covariates. The SAPALDIA cohort study procedures comply with the Declaration of Helsinki. For each survey, ethics approvals were granted by the regional ethics committees and participants provided written informed consent prior to participation.

## 2.2 Measures of Health-Related Quality Of Life (HRQoL)

The SAPALDIA 4 questionnaires included the 36-Item Short-Form Health Survey (SF-36), a widely-used and validated tool for measuring HRQoL in both population-based and clinical settings [31,32]. The questionnaire provides a summary of physical (PCS) and mental health (MCS) component scores, based on eight domains. The physical component comprises physical functioning (PF), bodily pain (BP), role physical (RP) and general health perception (GH). The mental component comprises vitality (VT), social role functioning (SF), role emotional (RE) and mental health perception (MH). Scores for each subscale range from 0–100, and higher scores indicate better HRQoL [33]. In our results we considered the two main domains GH & MH.

## 2.3 Measures of perceived built environment

We extracted relevant information on the perceived built from the SAPALDIA4 questionnaire. We considered personal satisfaction with apartment and neighborhood (score of four questions); proximity (in minutes) to supermarkets, local services, restaurants and cafés, public transportation services, sports facilities, parks and green spaces as well as quiet places; transportation noise annoyance (standardized rating Scale 0–10) [34].

## 2.4 Health care utilization

We defined health care utilization as use of medical services, also measured using the SAPALDIA4 questionnaire. We defined it as a variable, which combined the visit of either physician (s) or hospital(s) in the 12 months preceding the survey (0, 1 and 2 visits respectively).

## 2.5 Potential confounders

We *a priori* selected the following potential confounders measured at SAPALDIA4, based on existing literature and prior knowledge: age (years), sex (male/female), years of formal education ($\leq$9/$\leq$12/>12 years equivalent to primary, secondary and tertiary education), occupational status (full-time job, part-time job, retired, retired but still working); study area (Basel, Wald, Geneva, Payerne, Lugano, Aarau, Davos, Montana), smoking status (never/former/current).

   We specifically investigated the effect of additional adjustment for the social environment—living status of the participants (living alone vs. living with a partner) and social engagement (score built on eleven items); the specific questions are displayed in Table A1 in S1 Appendix.

   Moreover, we investigated the effect of additional adjustment for physical activity (sufficient moderate to vigorous physical activity (<150/$\geq$150 minutes per week)) and body mass index (BMI; kg/m$^2$).

## 2.6 Statistical analysis

In a first step (see 3.1), we described the characteristics of the study population, summarizing continuous variables as means and interquartile ranges (SF-36), and categorical variables as proportions. The median HRQoL GH score and the percentage of persons with at least one physician or hospital visit in the last 12 month are reported according to the levels of the characteristics.

   In a second step (see 3.2), we investigated associations of perceived built environment variables with HRQoL using multiple quantile regression models mutually adjusted for predictor variables while adjusting for covariates (sex, age, education, occupational status, smoking status and study area). We chose this approach as values of SF-36 derived HRQoL scores are

highly left-skewed, which means that most participants scored relatively high on the investigated scales (Figure A1 in S1 Appendix).

In a third step (see 3.3) we examined the modifying role of the social environment (living alone versus with a partner & social engagement) as well as physical activity and BMI in the association of the perceived built environment with HRQoL.

In a fourth step (see 3.4) we examined the associations of the perceived built environment with health care utilization, modified by the above mentioned variables, by performing multinomial (polytomous) logistic regression models.

We assessed all variables of the perceived built environment along their tertiles (low, medium and high). Due to their skewed distribution and the limited number of subjects in the respective categories, it was often not possible to have equal number of participants in each class as seen in Table 1. All of the above models were adjusted for potential individual-level and context-level confounders measured, including sex, age, education, occupational status, smoking status and study area.

We performed all analyses using Stata 15 (Stata Corporation, College Station, Texas) and considered associations as statistically significant at an alpha-level of 0.05. We conducted a total of 3 different statistical tests (not considering models that tested for the effect of additional adjustment). We provide in the footnote of the Tables information on which tests remained statistically significant after Bonferroni correction (adjusted p-values for 3 Models (General Health, Mental Health and Health care utilization).

## 3. Results

### 3.1 Characteristics of the study population

The characteristics of the study population are presented in Table 1. The mean age of the included participants was 64 years (43 to 87 years), with an equal distribution by sex. Approximately 61% of the subjects reported medium education levels. Half of the participants were still occupationally active (full-time or part-time) and half were retired. Relatively few participants were current smoker (15%) and nearly two third (64%) met the WHO guidelines for physical activity. 52% of the study population reported being satisfied with their apartment and neighborhood. With regards to perceived proximity measures, about a fourth of the study participants reported high levels of proximity to social places, sports facilities and quiet green places, whereas 55% reported public transportation to be available in proximity to their residence. Most subjects (75%) lived with a partner and showed low to medium social engagement.

On average participants reported high HRQoL scores across all domains. The median score of the GH HRQoL domain showed small or no differences by sex, proximity to social places, sports facilities and public transportation and peer support for daily activities. Descriptive differences in visits to either physicians and/or hospitals the last 12 months were detected for sex, age categories, noise annoyance ratings, occupational status, education and smoking status. The correlations between the social and perceived built environment variables are summarized in Table A2 in S1 Appendix.

### 3.2. Associations of perceived built environment with HRQoL

The results on the covariate adjusted associations of variables (categorized as tertiles) describing the perceived built environment with HRQoL domains are illustrated in Fig 1A and 1B. The middle tertile of self-reported satisfaction with the apartment and neighbourhood showed statistically significant positive associations with GH (4.09 (95%CI: 1.85; 6.34)), while the upper tertiles showed statistically significant positive associations with GH (5.49 (3.56; 7.42))

**Table 1. Characteristics of the study populations and sub-group specific HRQoL score (GH) and health care utilization.**

| Variable | Total n = 1980 | Percent (%) | Median score of overall HRQoL (GH) | Visited physician/hospital ≥1 previous 12 months (%) |
|---|---|---|---|---|
| Sex | | | | |
| Male | 1013 | 51 | 71 | 80 |
| Female | 967 | 49 | 72 | 89 |
| Age (Mean, SD) | 64.20(10.21) | | | |
| Age (years) | | | | |
| <55 | 1025 | 52 | 74 | 81 |
| 55–64 | 652 | 33 | 70 | 89 |
| ≥65 | 303 | 15 | 67 | 88 |
| Education | | | | |
| Low | 57 | 3 | 69 | 93 |
| Middle | 1209 | 61 | 72 | 84 |
| High | 714 | 36 | 72 | 84 |
| Occupational status | | | | |
| Full-time | 674 | 34 | 74 | 78 |
| Part-time | 294 | 15 | 74 | 86 |
| Retired | 758 | 38 | 68 | 89 |
| Retired & Working | 254 | 13 | 72 | 87 |
| Smoking Status | | | | |
| Never | 879 | 44 | 73 | 83 |
| Former | 812 | 41 | 70 | 87 |
| Current | 289 | 15 | 71 | 82 |
| Satisfaction with apartment and neighbourhood | | | | |
| Low | 469 | 24 | 66 | 86 |
| Medium | 471 | 24 | 72 | 83 |
| High | 1040 | 52 | 73 | 85 |
| Proximity to social places | | | | |
| Low | 677 | 34 | 71 | 83 |
| Medium | 780 | 39 | 72 | 85 |
| High | 523 | 26 | 71 | 87 |
| Proximity to public transportation | | | | |
| Low | 318 | 16 | 71 | 84 |
| Medium | 574 | 29 | 71 | 87 |
| High | 1088 | 55 | 72 | 83 |
| Proximity to sports facilities | | | | |
| Low | 841 | 43 | 71 | 85 |
| Medium | 583 | 30 | 72 | 85 |
| High | 555 | 28 | 71 | 83 |
| Proximity to quiet green places | | | | |
| Low | 732 | 37 | 70 | 85 |
| Medium | 658 | 33 | 72 | 84 |
| High | 590 | 30 | 73 | 84 |
| Noise annoyance | | | | |
| Low | 812 | 41 | 73 | 81 |
| Mid | 601 | 30 | 70 | 87 |
| High | 568 | 29 | 71 | 86 |
| Living alone | 498 | 25 | 70 | 85 |

*(Continued)*

**Table 1.** (Continued)

| Variable | Total n = 1980 | Percent (%) | Median score of overall HRQoL (GH) | Visited physician/hospital ≥1 previous 12 months (%) |
|---|---|---|---|---|
| Living with a partner | 1482 | 75 | 72 | 84 |
| Social engagement | | | | |
| Low | 728 | 37 | 70 | 84 |
| Medium | 670 | 34 | 72 | 84 |
| High | 582 | 29 | 73 | 86 |
| Physical Activity Guidelines (WHO) | | | | |
| Inactive | 709 | 36 | 67 | 84 |
| Sufficiently active | 1271 | 64 | 74 | 85 |
| BMI (Median) | 25.5 | | | |
| Physician/Hospital visit last 12 months | | | | |
| 0 | 307 | 16 | 77 | n.a |
| 1 | 1371 | 69 | 72 | n.a |
| 2+ | 302 | 15 | 65 | n.a |

Education: Low = Primary School (≤ 9years), Middle = Secondary school, middle school or apprenticeship (≤12 years), High = Technical College or University (≥12 years); Occupational status: Unemployment omitted due to class size (n = 11).

Physical Activity Guidelines (WHO).

Inactive: <150 min of MPA and <75 VPA per week.

Sufficient: >150 min of MPA or >75 VPA per week.

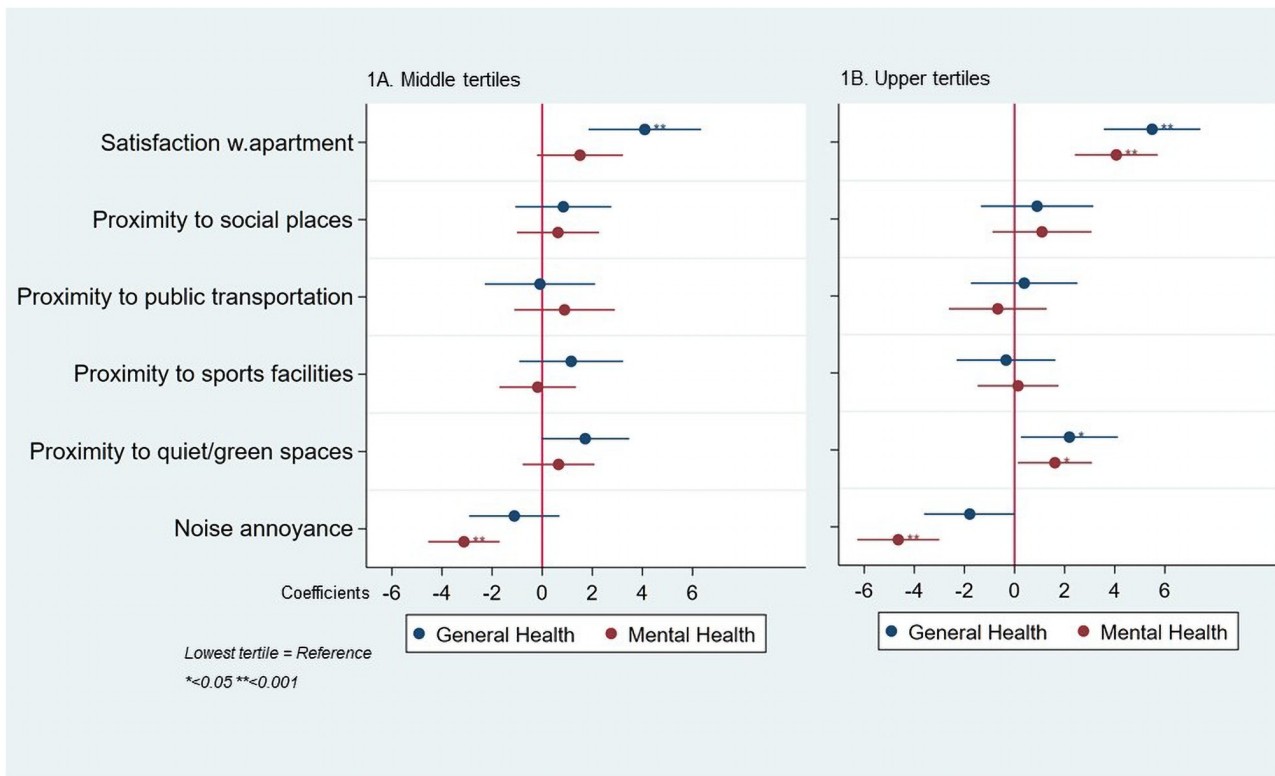

**Fig 1. Association of variables describing the perceived built environment categorized as tertiles (1A = Middle tertiles; 1B = Upper tertiles) with health-related quality of life domains, adjusted for covariates (sex, age, education, smoking status, occupational status and study area).**

and MH (4.07 (2.41; 5.72)), displaying a dose-response relationship. We found no association between proximity measures and HRQoL apart from a positive association of reported proximity to quiet and green spaces for the upper tertiles of this variable with both GH (1.61 (0.13; 3.09)) and MH (1.61 (0.13; 3.09)). We found a negative trend between tertiles of noise annoyance and HRQoL parameters. Compared to participants in the lowest tertile of noise annoyance, those in the middle and highest tertiles of noise annoyance showed statistically significant lower scores for MH (mid = -3.12 (-4.55; -1.70); high = -4.64 (-6.29; -3.00)). MH (mid = -3.37 (-4.81; -1.96); high = -4.57 (-6.15; -2.99)), with the highest tertile group having the lowest scores in this HRQoL parameters.

### 3.3 The role of adjusting for social environment, physical activity & BMI in the association of the perceived build environment with HRQoL

We observed no substantial differences in the association between the perceived built environment and HRQoL when adjusting the models for the social environment as well as for physical activity and BMI respectively (Table 2).

### 3.4 Perceived built environment and health care utilization

The results of the covariate adjusted associations of the perceived built environment variables with health care utilization are shown in Table 3. Participants reporting closer proximity to social places showed an increase of health care utilization with a relative risk ratio (RRR) of 1.54 (95%CI: 1.04; 2.39) compared to participants reporting living distant from social places. Subjects in the upper tertile of living proximate to sports facilities showed a decreased relative risk of visiting either physicians or hospitals more than once a year RRR = 0.56 (0.35; 0.88). We observed positive associations of noise annoyance with health care utilization for subjects in the middle tertile with 1 visit (RRR = 1.44 (1.05; 1.97)) and subjects in the upper tertile with more than 2 visits (RRR = 1.55 (1.00; 2.39)). When adjusting for the social environment, physical activity and BMI we did not observe substantial differences in the above mentioned associations.

## 4. Discussion

The results of this study are in agreement with a beneficial effect on general and mental HRQoL of satisfaction with one's apartment and the built environment around the residence and of proximity to green space. Only in the case of noise annoyance, which was associated with decreased HRQoL, did this association extend to an increased health care utilization. Proximity to social places was also associated with increased health care utilization, whereas proximity to sports facilities was associated with decreased health care utilization. Adjustment for the social environment or for physical activity and BMI did not change any of the associations.

A significant component of the perceived built environment was satisfaction with the apartment and neighbourhood. This variable associated most strongly with higher scores in both measured HRQoL domains. The results were consistent after adjusting for variables describing the social environment. The findings of Wong et al., 2018 agree with our results, even though the study was conducted in a cultural and geographical different region (Hong Kong) and with somewhat younger populations (on average 45 years) compared to the current study [35].

The observation of higher noise annoyance being associated with poorer HRQoL, especially for MCS, agrees with similar findings from several previous studies [36–39]. In addition, we observed a tendency of noise annoyance being associated with GH, suggesting that there might be an influence on poorer HRQoL aspects related to PCS. These findings not only add

**Table 2. Alteration in associations of variables defining the perceived built environment with health-related quality of life by adjustment of social environment variables, physical activity and BMI.**

| Perceived built environment | General Health | Mental Health |
|---|---|---|
| Ref = Lowest tertile | Coef (95% CI) | Coef (95% CI) |
| Satisfaction with Apartment and Built Environment | | |
| Mid tertile | 4.09 (1.85; 6.34)** + | 1.51 (-0.21; 3.22) |
| Upper tertile | 5.49 (3.56; 7.42)** + | 4.07 (2.41; 5.72)** + |
| Proximity to social places | | |
| Mid tertile | 0.84 (-1.08; 2.76) | 0.63 (-1.01; 2.27) |
| Upper tertile | 0.90 (-1.34; 3.14) | 1.10 (-0.88; 3.07) |
| Proximity to public transportation | | |
| Mid tertile | -0.09 (-2.29; 2.11) | 0.89 (-1.12; 2.89) |
| Upper tertile | 0.39 (-1.74; 2.92) | -0.67 (-2.62; 1.29) |
| Proximity to sports facilities | | |
| Mid tertile | 1.16 (-0.92; 3.23) | -0.18 (-1.71; 1.35) |
| Upper tertile | -0.33 (-2.31; 1.64) | 0.14 (-1.48; 1.76) |
| Proximity to quiet green places | | |
| Mid tertile | 1.72 (-0.04; 3.47) | 0.65 (-0.78; 2.09) |
| Upper tertile | 2.19 (0.25; 4.14)* | 1.61 (0.13; 3.09)* |
| Noise annoyance | | |
| Mid tertile | -1.11 (-2.92; 0.65) | -3.12 (-4.55; -1.70)** + |
| Upper tertile | -1.79; (-3.61; 0.03) | -4.64 (-6.29; -3.00)** + |
| *+ Social environment (Living alone versus with a partner & social engagement)* | | |
| Satisfaction with Apartment and Built Environment | | |
| Mid tertile | 4.56 (2.96; 6.63)** + | 0.84 (-0.84; 2.52) |
| Upper tertile | 5.97 (4.14; 7.80)** + | 3.85 (2.19; 5.50)** + |
| Proximity to social places | | |
| Mid tertile | 1.25 (-0.60; 3.10) | 1.01 (-0.56; 2.59) |
| Upper tertile | 1.16 (-1.13; 3.45) | 1.48 (-0.49; 3.46) |
| Proximity to public transportation | | |
| Mid tertile | -0.33 (-2.55; 1.89) | 1.00 (-0.94; 2.94) |
| Upper tertile | 0.65 (-1.56; 2.85) | 0.08 (-1.90; 1.73) |
| Proximity to sports facilities | | |
| Mid tertile | 0.93 (-1.07; 2.93) | -0.53 (-1.99; 0.93) |
| Upper tertile | -0.20 (-2.13; 1.74) | -0.69 (-2.22; 0.83) |
| Proximity to quiet green places | | |
| Mid tertile | 1.46 (-0.25; 3.18) | 0.61 (-0.77; 1.99) |
| Upper tertile | 1.80 (-0.10; 3.71) | 1.72 (0.27; 3.17)* |
| Noise annoyance | | |
| Mid tertile | -1.02 (-2.85; 0.82) | -3.51 (-4.82; -2.20)** + |
| Upper tertile | -1.62 (-3.39; 0.13) | -4.22 (-5.91; -2.55)** + |
| *+ Physical Activity & BMI (without social characteristics)* | | |
| Satisfaction with Apartment and Built Environment | | |
| Mid tertile | 4.44 (2.56; 6.40)** + | 1.32 (-0.54; 3.18) |
| Upper tertile | 6.63 (4.93; 8.32)** + | 3.80 (2.11; 5.48)** + |
| Proximity to social places | | |
| Mid tertile | 0.75 (-0.91; 2.41) | 0.24 (-1.30; 1.78) |
| Upper tertile | -0.48 (-2.41; 1.45) | 0.72 (-1.22; 2.67) |
| Proximity to public transportation | | |

*(Continued)*

**Table 2.** (Continued)

| Perceived built environment | General Health | Mental Health |
|---|---|---|
| **Ref = Lowest tertile** | **Coef (95% CI)** | **Coef (95% CI)** |
| Mid tertile | 0.39 (-1.98; 2.75) | 1.05 (-0.80; 2.90) |
| Upper tertile | 1.50 (-0.73; 3.72) | -0.19 (-1.94; 1.57) |
| Proximity to sports facilities | | |
| Mid tertile | 0.83 (-1.07; 2.72) | 0.14 (-1.65; 1.38) |
| Upper tertile | -0.10 (-1.52; 1.71) | -0.45 (-1.26; 1.26) |
| Proximity to quiet green places | | |
| Mid tertile | 1.07 (-0.36; 2.51) | 1.05 (-0.44; 2.54) |
| Upper tertile | 1.18 (-0,48; 2.85) | 2.32 (0.83; 3.81)* |
| Noise annoyance | | |
| Mid tertile | -1.96 (-3.49; -0.43)* + | -3.75 (-5.16; -2.35)** + |
| Upper tertile | -2.07 (-3.65; -0.48)* + | -4.37 (-5.98; -2.76)** + |

*p<0.05

**p<0.001;

+p<0.05 after Bonferroni correction.

Results were calculated using multivariate quantile regression model mutually adjusted for all exposure variables and confounders.

HRQoL was assessed using the SF-36.

Confounders: Sex, age, education, occupational status, smoking status, study area.

to the amount of literature showing adverse health effects of noise annoyance [40–42], but go a step further in showing increased need of healthcare and use of medical services for individuals reporting high noise annoyance ratings.

Our findings indicate that the perceived proximity to cultural, sports as well as public transportation may not be major determinants of HRQoL. Regarding these proximity measures, our results contradict some studies [12,43,44], yet agree with another study, assessing 5000 adults in Berlin, Paris, London, New York and Toronto, which suggests no direct association of neighbourhood proximity characteristics with HRQoL for older adults (similar age distribution as this study) [45]. On the contrary, the same study found relevant association of proximity measures for younger adults and declared that older adults valued provision of services and healthcare facilities more, compared to proximity to social and recreational amenities. There might be several explanations for the lack of associations with proximity characteristics. Residents with very low HRQoL could be less aware of a city's attractiveness as they leave their apartment less frequently. A hypothesis of Machón et al. 2017 stated that if people live for many decades in the same city they get used to the environment, which could lead to a lack of associations with HRQoL [46]. A possible approach to overcome these issues and increase HRQoL of city residents is communal living. This type of living environment is expected to improve the housing crisis and at the same time help people in need, such as disabled older aged persons [47].

However, we can only hypothesize about these clarifications, as there may be numerous unknown factors contributing to individual preferences or aversions when dealing with perceptions of environments. Also, the cross-sectional nature of the study does not allow investigation in the directionality of the associations.

Regarding health care utilization, noise annoyance showed statistically significant associations with visiting physicians or hospitals more than once a year. This implies that the

**Table 3. Associations of variables defining the perceived built environment with health care utilization, with and without adjustment for the social environment, physical activity and BMI.**

| Perceived built environment | Combined (physician & hospital) RRR (95% CI) | | |
|---|---|---|---|
| | 0 = Reference | 1 | >2 |
| Satisfaction with Apartment and Built Environment | | | |
| Mid tertile | | 0.91 (0.63; 1.33) | 1.24 (0.76; 2.03) |
| Upper tertile | | 1.01 (0.73; 1.41) | 1.23 (0.79; 1.90) |
| Proximity to social places | | | |
| Mid tertile | | 1.12 (0.81; 1.55) | 1.23 (0.81; 1.88) |
| Upper tertile | | 1.54 (1.04; 2.39)* | 1.63 (0.99; 2.68) |
| Proximity to public transportation | | | |
| Mid tertile | | 1.24 (0.81; 1.91) | 1.62 (0.91; 2.89) |
| Upper tertile | | 0.89 (0.60; 1.33) | 1.25 (0.73; 2.17) |
| Proximity to sports facilities | | | |
| Mid tertile | | 0.96 (0.68; 1.35) | 0.88 (0.57; 1.36) |
| Upper tertile | | 0.88 (0.62; 1.25) | 0.56 (0.35; 0.88)* + |
| Proximity to quiet green places | | | |
| Mid tertile | | 0.98 (0.72; 1.34) | 1.06 (0.72; 1.58) |
| Upper tertile | | 1.11 (0.80; 1.52) | 1.07 (0.71; 1.63) |
| Noise annoyance | | | |
| Mid tertile | | 1.44 (1.05; 1.97)* | 1.26 (0.84; 1.89) |
| Upper tertile | | 1.39 (1.00; 1.94) | 1.55 (1.00; 2.39)* |
| **+ Social environment (Living alone versus with a partner & social engagement)** | | | |
| Satisfaction with Apartment and Built Environment | | | |
| Mid tertile | | 0.89 (0.61; 1.29) | 1.22 (0.75; 1.99) |
| Upper tertile | | 0.98 (0.70; 1.37) | 1.21 (0.77; 1.87) |
| Proximity to social places | | | |
| Mid tertile | | 1.15 (0.83; 1.59) | 1.25 (0.82; 1.91) |
| Upper tertile | | 1.61 (1.08; 2.40)* | 1.67 (1.02; 2.75)* |
| Proximity to public transportation | | | |
| Mid tertile | | 1.26 (0.82; 1.94) | 1.64 (0.92; 2.92) |
| Upper tertile | | 0.90 (0.60; 1.35) | 1.27 (0.73; 2.19) |
| Proximity to sports facilities | | | |
| Mid tertile | | 0.94 (0.67; 1.31) | 0.86 (0.56; 1.33) |
| Upper tertile | | 0.86 (0.60; 1.22) | 0.54 (0.344; 086)* + |
| Proximity to quiet green places | | | |
| Mid tertile | | 0.98 (0.71; 1.34) | 1.06 (0.72; 1.58) |
| Upper tertile | | 1.09 (0.78; 1.52) | 1.07 (0.71; 1.62) |
| Noise annoyance | | | |
| Mid tertile | | 1,43 (1.04; 1.95)* | 1.26 (0.84; 1.90) |
| Upper tertile | | 1.41 (1.01; 1.97)* | 1.56 (1.01; 2.42)* |
| **+ Physical Activity & BMI (without social engagement)** | | | |
| Satisfaction with Apartment and Built Environment | | | |
| Mid tertile | | 0.94 (0.64; 1.37) | 1.36 (0.83; 2.22) |
| Upper tertile | | 1.00 (0.72; 1.40) | 1.26 (0.81; 1.96) |
| Proximity to social places | | | |
| Mid tertile | | 1.12 (0.81; 1.55) | 1.23 (0.80; 1.87) |
| Upper tertile | | 1.58 (1.06; 2.35)* | 1.64 (0.99; 2.69) |
| Proximity to public transportation | | | |

*(Continued)*

**Table 3.** (Continued)

| Perceived built environment | Combined (physician & hospital) RRR (95% CI) | | |
|---|---|---|---|
| | 0 = Reference | 1 | >2 |
| Mid tertile | | 1.24 (0.80; 1.91) | 1.62 (0.91; 2.90) |
| Upper tertile | | 0.90 (0.60; 1.34) | 1.27 (0.73; 2.21) |
| Proximity to sports facilities | | | |
| Mid tertile | | 0.96 (0.68; 1.35) | 0.93 (0.60; 1.44) |
| Upper tertile | | 0.87 (0.61; 1.24) | 0.58 (0.37; 0.91)* |
| Proximity to quiet green places | | | |
| Mid tertile | | 1.00 (0.73; 1.37) | 1.10 (0.74; 1.63) |
| Upper tertile | | 1.14 (0.82; 1.58) | 1.11 (0.73; 1.69) |
| Noise annoyance | | | |
| Mid tertile | | 1.49 (1.09; 2.04)* + | 1.36 (0.91; 2.05) |
| Upper tertile | | 1.46 (1.04; 2.04)* | 1.73 (1.12; 2.69)* + |

*$p < 0.05$

**$p < 0.001$;

+$p < 0.05$ after Bonferroni correction.

RRR = Relative risk ratios.

Results were calculated using multinomial (polytomous) logistic regression models mutually adjusted for all exposure variables and confounders.

Physician and hospital visits were self-reported for the last 12 months.

Confounders: Sex, age, education, occupational status, smoking status, study area.

association of transportation noise annoyance with HRQoL has downstream costs by leading to increased doctors and hospital visits. We further identified an increased use of health services for people living closer to social places and a decreased use for people living closer to sports facilities. These findings may imply a connection of living closer to social places, and most importantly medical facilities with an increased use of health services. In contrast, living closer to sports facilities may be one of many factors that prevents an increased use of health services. Future studies need to investigate the cost-effectiveness of decreasing transportation noise in urban environments and further investigate the associations of living closer to sports facilities and social places with health service utilization.

As we did not find substantial differences when adjusting for social environment, physical activity and BMI, independent pathways from the built environment to HRQoL and health care utilization may be expected and need to be investigated in future studies.

## 4.1 Strength and limitations

A major strength of this study is the comprehensive consideration of the perception of built environmental parameters with HRQoL outcomes and healthcare seeking behavior. Exhaustive analysis were conducted to investigate independence of these associations from the social environment and lifestyle behavior. In addition, the investigation with health care utilization, facilitates the transfer of our results to clinical relevant domains plus builds a basis for health economic evaluation of environmental risks and burdens for healthcare systems. The population-based design of this study favors the generalizability of the findings within the Swiss setting. However, due to participation and survivor bias, validity and generalizability are always at risk in longitudinal cohort settings. In particular, compared to similar settings the sample from Switzerland aged 55 years and older, showed higher HRQoL scores, which may be an issue when comparing with other countries and studies.

Due to the cross-sectional nature of the study, inferring causality and directionality of the associations is not possible. This may be particularly relevant for the observed association of proximity to social places with health care utilization. Persons with existing limitations and higher needs for health care services may choose to live closer to such services. We looked at perceptions of the built environment, which may have introduced a bias of subjective validation. However, the perception of environment is a relevant aspects despite being subjectively biased by nature.

Due to the lack of air pollution information at SAPALDIA 4, we were unable to take this potentially important confounder into consideration. Finally, due to the restricted sample size there is a chance that some relevant associations went unnoticed.

## 5. Conclusion

Our study contributes to the understanding of an independent role of the perceived built environment on residents' HRQoL. In particular, the study points to a potentially high benefit gained from decreasing transportation noise for both, HRQoL and health care utilization.

## Supporting information

**S1 Appendix. Supplementary information.**
(DOCX)

## Author Contributions

**Conceptualization:** Paco Cerletti, Ikenna C. Eze, Nicole Probst-Hensch.

**Data curation:** Dirk Keidel, Emmanuel Schaffner.

**Formal analysis:** Paco Cerletti, Dirk Keidel.

**Funding acquisition:** Nicole Probst-Hensch.

**Investigation:** Paco Cerletti, Ikenna C. Eze, Daiana Stolz, Paola M. Gasche-Soccal, Thomas Rothe, Medea Imboden, Nicole Probst-Hensch.

**Methodology:** Paco Cerletti, Ikenna C. Eze, Dirk Keidel, Medea Imboden, Nicole Probst-Hensch.

**Project administration:** Medea Imboden, Nicole Probst-Hensch.

**Supervision:** Nicole Probst-Hensch.

**Writing – original draft:** Paco Cerletti.

**Writing – review & editing:** Paco Cerletti, Ikenna C. Eze, Dirk Keidel, Emmanuel Schaffner, Daiana Stolz, Paola M. Gasche-Soccal, Thomas Rothe, Medea Imboden, Nicole Probst-Hensch.

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
