## [Decision Letter · Decision Letter 0]

16 Nov 2020

PONE-D-20-27261

Elucidating independent and joint associations of the social and perceived built environment with health-related quality of life and health care utilization

PLOS ONE

Dear Dr. Probst-Hensch,

Thank you for submitting your manuscript to PLOS ONE. After careful consideration, we feel that it has merit but does not fully meet PLOS ONE’s publication criteria as it currently stands. Therefore, we invite you to submit a revised version of the manuscript that addresses the points raised during the review process.

A **rebuttal letter** that responds to **EACH** point raised by the academic editor and reviewer(s). You should upload this letter as a separate file labeled 'Response to Reviewers'.A **marked-up copy** of your manuscript that highlights changes made to the original version. You should upload this as a separate file labeled 'Revised Manuscript with Track Changes'.An **unmarked version** of your revised paper without tracked changes. You should upload this as a separate file labeled 'Manuscript'.

We look forward to receiving your revised manuscript.

Kind regards,

Brecht Devleesschauwer

Academic Editor

PLOS ONE

Additional Editor Comments:

In your revision note, please include EACH of the reviewer comments, provide your reply, and when relevant, include the modified/new text (or motivate why you decided not to modify the text). Note that failure to do so may result in a rejection of the manuscript.

Journal Requirements:

Reviewers' comments:

Reviewer's Responses to Questions

**Comments to the Author**

1. Is the manuscript technically sound, and do the data support the conclusions?

Reviewer #1: Yes

Reviewer #2: Partly

2. Has the statistical analysis been performed appropriately and rigorously? 

Reviewer #1: I Don't Know

Reviewer #2: No

3. Have the authors made all data underlying the findings in their manuscript fully available?

Reviewer #1: No

Reviewer #2: No

4. Is the manuscript presented in an intelligible fashion and written in standard English?

Reviewer #1: Yes

Reviewer #2: Yes

5. Review Comments to the Author

Reviewer #1: This paper looks at associations between the built environment and health related quality of life

You have very few significant results among a lot of statistical tests – do you think your significant results might be type 1 error?

Line 40 what about the natural environment?

Line 42 define built and social environment here

Line 43 ‘impacts’ instead of ‘comprises’

Line4 64 define HRQOL and its relationship to health

Line 134 were those few values suggesting good, average or poor health?

Line 175 did you check for multicollinearity in your analysis? Did you account for sample design (clustering) in your analysis

Line 214 why use some tertiles and some dichotomised?

Line 226 – age at what point –baseline or final follow up

Line 229 are these characteristics typical for the Swiss population?

Discussion – Does the Swiss population have higher HRQOL than elsewhere and or higher housing quality than elsewhere? If so research elsewhere might find different associations?

Would you suggest communal living as a solution to living alone? https://www.theguardian.com/cities/2019/sep/03/co-living-the-end-of-urban-loneliness-or-cynical-corporate-dormitories

Reviewer #2: In general, the manuscript is very long and lack of conciseness. We fail to clearly understand the results and the key messages among the multitude of statistical analyses performed.

Regarding the research question, the scientific relevance of studying the joint association of the built and the social environment on HRQL/Health care utilization is not sufficiently explained. In the introduction, it is mentioned that the mechanisms are not yet well understood but the performed analysis do not allow to explore the mechanisms. The analysis with latent classes assess the association of the combined exposure to social and built environment with HQLR but not explain the mechanisms by which this environment affect the HRQL/Health care utilization. Mediation analyses or different regression models performed with an increasing level of adjustment for covariates would have been more adequate if the goal was to explore the mechanisms. It would have been interesting to consider the “social environment” as a mediator in the association between the built environment and the HRQL/health care utilization The ability of cities to offer places and services could affect the HRQL through the impact on social connectedness.

Therefore, you should be careful with the following sentence in your conclusion: “ our study adds to the understanding of how the social and built environment contributes to the HRQL”. I would rather talk about the characteristics of the social and built environment that are associated with HRQL.

Question: what is the main added value of the analysis performed with the Latent classes? In my view, it could have been interesting to use them in case of high multi-collinearity between variables or if latent classes would have provided very distinctive groups. If the aim of the study is to assess the joint association, why didn’t you choose instead to include interaction terms in the regression model?

Also, including each dimension of the HRQL in the analyses (BP, VT, etc.) seems to me too exhaustive and make the final interpretation more complicated. The interpretation could have been simplified by using an aggregated score of the HRQL.

The performed statistical models and the data do not allow to completely answer the research question which aim to investigate the independent and joint association between the perceived built and social environment and HRQL and Health care utilization. All your models are adjusted for lifestyle factors (physical activity, BMI and smoking) that are potential mediators in the association between the built environment and health. The estimates of your model do not take into account the part of the effect due to those factors. However, we know from the literature that one of the mechanism by which the built environment affect the health status is linked to healthy lifestyles. For example, many urban environments lack green and open spaces that encourage exercise. In your discussion part (line 449), you write “ the perceived proximity to cultural, sports and green amenities may not be a major determinant of HRQL”, but this result is maybe due to the fact that you adjusted your model for “physical activity”.

Some comments regarding the validity of the variables selected in the models to approach the concept of social environment, built environment and Health care utilization:

The social environment is a very wide concept that encompasses many dimensions such as social support, socio-economic status, family composition etc. The distinction between the variables selected to define the “social environment” and the socio-economic variables selected to adjust the model is not clear. Why did you choose to include the “occupational status” in the “social environment” and not in the confounders such as the “educational level”? Regarding the “occupational status”, why is there no unemployment status?

An important dimension of the built environment that is missing in your study is air pollution. This should be mentioned in the discussion. Air pollution could be a confounding factor in the association found between noise annoyance and HRQL.

Regarding the health care utilization, you decide to dichotomize the variable, with 0 or >=1 Physician visit in the last 12 months. But what do we really aim to measure with this binomial variable? What is considered as a normal or healthy behavior in health care utilization? Is the variable really measuring a poor health status or could it reflect a problem of access to health care?

Specific comments:

Line 540: “an increased satisfaction living situation associated with increased health care utilization”. I do not find this association in the results.

Lline 583: A point is lacking

Table 2: you could make a smaller table by removing one of the two lines describing the variables with 2 categories (High/low)

Figure 3: in my view, this figure could be in the annexes

Recommendation:

- Run the models without adjusting for BMI and Physical activity

6. PLOS authors have the option to publish the peer review history of their article (what does this mean?). If published, this will include your full peer review and any attached files.

Reviewer #1: No

Reviewer #2: No

---

## [Author Response · Author response to Decision Letter 0]

9 Mar 2021

Dear Reviewers,

Many thanks for your time and very constructive comments. We substantially changed the focus of the manuscript on the basis of your comments. We redirected the aim of the manuscript to investigating the association of the perceived built environment with health-related quality of life and health care utilization. Moreover, we elucidated the independence of the observed associations from social environment, physical activity and BMI.

Please find all the specific changes and answers to your comments below.

Reviewer #1: This paper looks at associations between the built environment and health related quality of life

You have very few significant results among a lot of statistical tests – do you think your significant results might be type 1 error?

- We now adapted the objectives of this manuscript as described above. Furthermore, we only considered the two main domains of the SF-36 (GH & MH) to generate more concise and concrete results. The perceived environmental domains were mutually adjusted in the models, however we ran separate models for the associations with HRQoL and health care utilization and conducted a total of 3 statistical tests (not considering the additional models testing the effect of additional adjustment). We therefore provide in the footnote of each table the Bonferroni corrected results.

Line 40 what about the natural environment?

- We added the natural environment (line 47)

Line 42 define built and social environment here

- Definition of the built environment have been added. Social environment adapted as not anymore the focus of the manuscript (line 47-54). 

Line 43 ‘impacts’ instead of ‘comprises’

- Changed accordingly (line 50)

Line4 64 define HRQOL and its relationship to health

- We added an information on the correlation (line 60-61)

Line 134 were those few values suggesting good, average or poor health?

- We changed this section and added explanations. 

Line 175 did you check for multicollinearity in your analysis? Did you account for sample design (clustering) in your analysis

- We did not detect high correlations between the predictor variables as seen in Table A1.

Line 214 why use some tertiles and some dichotomised?

- We now consistently used tertiles for all the predictor variables.

Line 226 – age at what point –baseline or final follow up

- We added the information that data was captured at SAPALDIA4 in section 2.1.

Line 229 are these characteristics typical for the Swiss population?

- We excluded the latent class analysis, hence this information is not listed anymore. 

Discussion – Does the Swiss population have higher HRQOL than elsewhere and or higher housing quality than elsewhere? If so research elsewhere might find different associations?

- We added information on this in the limitations section of the discussion

Would you suggest communal living as a solution to living alone? https://www.theguardian.com/cities/2019/sep/03/co-living-the-end-of-urban-loneliness-or-cynical-corporate-dormitories

- Indeed a very interesting approach, we added information on communal living in the discussion (line 338-341)

Reviewer #2: In general, the manuscript is very long and lack of conciseness. We fail to clearly understand the results and the key messages among the multitude of statistical analyses performed.

Regarding the research question, the scientific relevance of studying the joint association of the built and the social environment on HRQL/Health care utilization is not sufficiently explained. In the introduction, it is mentioned that the mechanisms are not yet well understood but the performed analysis do not allow to explore the mechanisms. 

The analysis with latent classes assess the association of the combined exposure to social and built environment with HQLR but not explain the mechanisms by which this environment affect the HRQL/Health care utilization. Mediation analyses or different regression models performed with an increasing level of adjustment for covariates would have been more adequate if the goal was to explore the mechanisms. It would have been interesting to consider the “social environment” as a mediator in the association between the built environment and the HRQL/health care utilization The ability of cities to offer places and services could affect the HRQL through the impact on social connectedness.

Therefore, you should be careful with the following sentence in your conclusion: “ our study adds to the understanding of how the social and built environment contributes to the HRQL”. I would rather talk about the characteristics of the social and built environment that are associated with HRQL.

- We thank the reviewer for these helpful and important comments. It motivated us to reshape the objectives of the paper. Given the cross-sectional nature of our study, we abstained from conducting a mediation analysis. But we investigated whether or not the association of the perceived built environment is altered by adjustment for the social environment and for physical activity or BMI. As mentioned above we now redirected the objectives of the manuscript to focusing on the association of the perceived built environment with HRQoL and health care utilization.

Question: what is the main added value of the analysis performed with the Latent classes? In my view, it could have been interesting to use them in case of high multi-collinearity between variables or if latent classes would have provided very distinctive groups. If the aim of the study is to assess the joint association, why didn’t you choose instead to include interaction terms in the regression model?

- We excluded the LCA as we agree on the poor added value. 

Also, including each dimension of the HRQL in the analyses (BP, VT, etc.) seems to me too exhaustive and make the final interpretation more complicated. The interpretation could have been simplified by using an aggregated score of the HRQL.

- We do agree also on this point and included only GH and MH in the results.

The performed statistical models and the data do not allow to completely answer the research question which aim to investigate the independent and joint association between the perceived built and social environment and HRQL and Health care utilization. All your models are adjusted for lifestyle factors (physical activity, BMI and smoking) that are potential mediators in the association between the built environment and health. The estimates of your model do not take into account the part of the effect due to those factors. However, we know from the literature that one of the mechanism by which the built environment affect the health status is linked to healthy lifestyles. For example, many urban environments lack green and open spaces that encourage exercise. In your discussion part (line 449), you write “ the perceived proximity to cultural, sports and green amenities may not be a major determinant of HRQL”, but this result is maybe due to the fact that you adjusted your model for “physical activity”.

- Please see our comments above on the redirection of the manuscript.

Some comments regarding the validity of the variables selected in the models to approach the concept of social environment, built environment and Health care utilization:

The social environment is a very wide concept that encompasses many dimensions such as social support, socio-economic status, family composition etc. The distinction between the variables selected to define the “social environment” and the socio-economic variables selected to adjust the model is not clear. Why did you choose to include the “occupational status” in the “social environment” and not in the confounders such as the “educational level”? Regarding the “occupational status”, why is there no unemployment status?

- We now included occupational status as a confounder. Unemployment was excluded as the group size was very small (n=11) and would have distorted the results. We now show the specific impact of adjusting for the social variables “social engagement” and “living alone”.

An important dimension of the built environment that is missing in your study is air pollution. This should be mentioned in the discussion. Air pollution could be a confounding factor in the association found between noise annoyance and HRQL.

- We agree. However, we could not include air pollution, as it was not measured in the last follow-up of SAPALDIA. We included this information in the limitations. 

Regarding the health care utilization, you decide to dichotomize the variable, with 0 or >=1 Physician visit in the last 12 months. But what do we really aim to measure with this binomial variable? What is considered as a normal or healthy behavior in health care utilization? Is the variable really measuring a poor health status or could it reflect a problem of access to health care?

- We now categorized the variable to 0, 1 2+ visits. 

Specific comments:

Line 540: “an increased satisfaction living situation associated with increased health care utilization”. I do not find this association in the results.

Line 583: A point is lacking

Table 2: you could make a smaller table by removing one of the two lines describing the variables with 2 categories (High/low)

Figure 3: in my view, this figure could be in the annexes

- As we substantially changed the manuscripts these comments were resolved.

Recommendation:

- Run the models without adjusting for BMI and Physical activity

- We ran the model with and without adjusting for physical activity and BMI

---

## [Decision Letter · Decision Letter 1]

6 Apr 2021

PONE-D-20-27261R1

Perceived built environment, health-related quality of life and health care utilization

PLOS ONE

Dear Dr. Probst-Hensch,

Thank you for submitting your manuscript to PLOS ONE. After careful consideration, we feel that it has merit but does not fully meet PLOS ONE’s publication criteria as it currently stands. Therefore, we invite you to submit a revised version of the manuscript that addresses the points raised during the review process.

We look forward to receiving your revised manuscript.

Kind regards,

Brecht Devleesschauwer

Academic Editor

PLOS ONE

Journal Requirements:

Reviewers' comments:

Reviewer's Responses to Questions

**Comments to the Author**

1. If the authors have adequately addressed your comments raised in a previous round of review and you feel that this manuscript is now acceptable for publication, you may indicate that here to bypass the “Comments to the Author” section, enter your conflict of interest statement in the “Confidential to Editor” section, and submit your "Accept" recommendation.

Reviewer #1: (No Response)

2. Is the manuscript technically sound, and do the data support the conclusions?

Reviewer #1: Yes

3. Has the statistical analysis been performed appropriately and rigorously? 

Reviewer #1: Yes

4. Have the authors made all data underlying the findings in their manuscript fully available?

Reviewer #1: No

5. Is the manuscript presented in an intelligible fashion and written in standard English?

Reviewer #1: No

6. Review Comments to the Author

Reviewer #1: This study looks at how health and wellbeing might be associated with the built environment

This study is much improved but I have a few comments:

Line 25 add ‘and health care utilisation’ after ‘(HRQoL)’ Change “. However” to “but”

Line 26-27 delete “and on how the HRQoL associations extend to health care utilization”

Lines 28-30 replace “the association of the perceived built environment with HRQoL and health care utilization, independent of the social environment, 30 physical activity and body mass index (BMI)” with “these associations” – otherwise very repetitive

Line 36-44 might be more clearly written e.g.

“Higher HRQOL and less health care utilisation were associated with less reported noise annoyance from transport. Higher HRQOL was also associated with greater satisfaction with the living environment and more perceived access to greenspaces. These results were independent of the social environment (living alone and social engagement) and lifestyle (physical activity level and BMI).

This study provides further evidence that the built environment should be designed to integrate living and green spaces but separate living and traffic spaces in order to improve health and wellbeing and potentially save health care costs.”

Line 47 Define natural, built, physical and social environments

Line 56 to 60 Combine these two studies into one sentence e.g. Health outcomes including headaches, arthritis , respiratory disease and obesity have been linked to the built environment (Goldbberg et al Hogan etal)

Line 60 define HRQOL

Line 66 to 69 I can’t follow this sentence – too long

Line 71-72 physical activity in relation to what ?

Line 72 to 74 – might be closer to healthcare in a town than in the countryside so I don’t undersand this. Also you don’t look at interactions with age so probably irrelevant for your study

Line 75 -77. Reverse causation here: depression might cause people to rate their support as poor

Line 84 do you mean physical limitations associated with obesity?

Line 95 your study does not include economic analysis of cost-benefit so I would rewrite this – instead talk about studies showing how HRQOL leads to health care utilisation and health care costs in Switzerland

Line 97 Either change “only very few” to “no studies that we could find” or describe the studies

Line 182 delete ‘at salpadia4’

Line 184 –explain further what this means (most respondents scored quite high on the scales??)

Line 195 – what do you mean by ‘classes’

Table 1 – is your study population similar to census characteristics of the Swiss population?

Figure 1 Please label the X axes

Line 299 word missing?

Line 301 why are these ‘critical’?

Line 303 delete ‘mutually’

Line 309 Perception being more important than actual built environment could suggest confounding by some sort of generalised negative affect so I would move that to the limitations section

Line 314 to 322 I think this paragraph relates to your old findings and should be deleted

Line 324 you did find sports was important

Line 325 to line 345 This paragraph looks at interactions with age which you did not explore so should be deleted. Instead you might need to explain why proximity to social spaces increased health care visits – it might be they were places that sell alcohol or unhealthy foods, perhaps to takeaway such as pizza?

Line 355 proximity to greenspace might have been confounded with physical activity and BMI?

Tables 2 and 3 – it would be useful to have bivariable associations in these tables

7. PLOS authors have the option to publish the peer review history of their article (what does this mean?). If published, this will include your full peer review and any attached files.

Reviewer #1: No

---

## [Author Response · Author response to Decision Letter 1]

19 Apr 2021

Dear Reviewers,

Many thanks for your time and specific comments. Please find all the specific changes and answers to your comments below.

Reviewer #1: This study looks at how health and wellbeing might be associated with the built environment

This study is much improved but I have a few comments:

Line 25 add ‘and health care utilisation’ after ‘(HRQoL)’ Change “. However” to “but”

• Adapted accordingly

Line 26-27 delete “and on how the HRQoL associations extend to health care utilization”

• Adapted accordingly

Lines 28-30 replace “the association of the perceived built environment with HRQoL and health care utilization, independent of the social environment, 30 physical activity and body mass index (BMI)” with “these associations” – otherwise very repetitive

• We replaced the first part with “these associations” but kept the second part, to specify, which lifestyle variables we investigated. 

Line 36-44 might be more clearly written e.g.

“Higher HRQOL and less health care utilisation were associated with less reported noise annoyance from transport. Higher HRQOL was also associated with greater satisfaction with the living environment and more perceived access to greenspaces. These results were independent of the social environment (living alone and social engagement) and lifestyle (physical activity level and BMI).

This study provides further evidence that the built environment should be designed to integrate living and green spaces but separate living and traffic spaces in order to improve health and wellbeing and potentially save health care costs.”

• Adapted accordingly. Except a minor change of “noise annoyance from transport” to “transportation noise annoyance”

Line 47 Define natural, built, physical and social environments

• Added definitions

Line 56 to 60 Combine these two studies into one sentence e.g. Health outcomes including headaches, arthritis, respiratory disease and obesity have been linked to the built environment (Goldbberg et al Hogan etal)

• We summarized the two references into one sentence

Line 60 define HRQOL

• Definition added

Line 66 to 69 I can’t follow this sentence – too long

• Sentence shortened and made more precise

Line 71-72 physical activity in relation to what ?

• Added “in relation to the built environment”

Line 72 to 74 – might be closer to healthcare in a town than in the countryside so I don’t undersand this. Also you don’t look at interactions with age so probably irrelevant for your study

• Deleted this sections as we agree that it is irrelevant for the new findings

Line 75 -77. Reverse causation here: depression might cause people to rate their support as poor

• Added a statement 

Line 84 do you mean physical limitations associated with obesity?

• Changed to physical activity limitations, to clarify the relation to the built environment.

Line 95 your study does not include economic analysis of cost-benefit so I would rewrite this – instead talk about studies showing how HRQOL leads to health care utilisation and health care costs in Switzerland

• Adapted this section and focused more on the objectives of the present study

Line 97 Either change “only very few” to “no studies that we could find” or describe the studies

• Adapted accordingly

Line 182 delete ‘at salpadia4’

• Adapted accordingly

Line 184 –explain further what this means (most respondents scored quite high on the scales??)

• Added a further explanation

Line 195 – what do you mean by ‘classes’

• Changed to “subjects in the respective categories”

Table 1 – is your study population similar to census characteristics of the Swiss population?

• We compared our study population, and hence the characteristics, to other populations in section 4.1

Figure 1 Please label the X axes

• Added a label 

Line 299 word missing?

• Added “change”

Line 301 why are these ‘critical’?

• Changed to significant

Line 303 delete ‘mutually’

• Adapted accordingly

Line 309 Perception being more important than actual built environment could suggest confounding by some sort of generalised negative affect so I would move that to the limitations section

• We deleted this section and added an explanation in the limitations section (Line 379)

Line 314 to 322 I think this paragraph relates to your old findings and should be deleted

• Dear Reviewer, we believe that the findings of noise annoyance on MCS are still very valid in our new findings. Hence, we would suggest to keep this section as it highlights that our findings on noise annoyance resonate well with other studies in this area.

Line 324 you did find sports was important

• In this line we refer to the findings for HRQoL, where we did not identify a relevant association of proximity to sports facilities with HRQoL. 

Line 325 to line 345 This paragraph looks at interactions with age which you did not explore so should be deleted. Instead you might need to explain why proximity to social spaces increased health care visits – it might be they were places that sell alcohol or unhealthy foods, perhaps to takeaway such as pizza?

• This section refers to the HRQoL findings. However, we added interpretations of the findings on proximity to social places and sports facilities in the next section. 

Line 355 proximity to greenspace might have been confounded with physical activity and BMI?

Tables 2 and 3 – it would be useful to have bivariable associations in these tables

• As we included Physical Activity and BMI in the models we adjusted for possible confounding of these variables in the association of green spaces and HRQoL. We showed that this association did not change substantially when including or excluding physical activity and BMI.

---

## [Editor Report · Decision Letter 2]

23 Apr 2021

Perceived built environment, health-related quality of life and health care utilization

PONE-D-20-27261R2

Dear Dr. Probst-Hensch,

We’re pleased to inform you that your manuscript has been judged scientifically suitable for publication and will be formally accepted for publication once it meets all outstanding technical requirements.

Kind regards,

Brecht Devleesschauwer

Academic Editor

PLOS ONE
---

## [Editor Report · Acceptance letter]

27 Apr 2021

PONE-D-20-27261R2 

Perceived built environment, health-related quality of life and health care utilization 

Dear Dr. Probst-Hensch:

I'm pleased to inform you that your manuscript has been deemed suitable for publication in PLOS ONE. Congratulations! Your manuscript is now with our production department. 

Kind regards, 

on behalf of

Prof. Dr. Brecht Devleesschauwer 

Academic Editor

PLOS ONE